# Effectiveness of care coordination interventions delivered to stroke survivors in low and middle-income countries: Systematic review and meta-analysis protocol

Stephanopoulos Kofi Junior Osei[1,2], Anthony Danso-Appiah[1,2,3]*

1 Department of Epidemiology and Disease Control, School of Public Health, University of Ghana, Legon, Ghana, 2 Centre for Evidence Synthesis and Policy, University of Ghana, Legon, Accra, Ghana, 3 Africa Communities of Evidence Synthesis and Translation, Accra, Ghana

* adanso-appiah@ug.edu.gh, tdappiah@yahoo.co.uk

## Abstract

### Background

Stroke survivors have complex and long-term care needs requiring navigation of multiple care services and providers. Care coordination interventions provide wholistic care that meets the needs of the patient and improves their clinical outcomes and expectations. This systematic review will identify the key components of stroke care coordination interventions implemented in low and middle-income countries (LMICs) and assess their effectiveness on stroke outcomes such as motor recovery, cognitive function, mental health, stroke type and the role of stroke severity, type, and the nature of the interventions in influencing these outcomes.

### Methods

Electronic databases, trial registries and non-database sources will be searched for published and unpublished studies. PubMed, LILACS, CINAHL via EBSCOhost, Scopus, Web of Science Core Collection, Cochrane CENTRAL and Google Scholar will be searched from 2000 to 31st May 2025, without language restriction. Trial registries including the WHO International Clinical Trials Registry Platform (ICTRP) and Clinicaltrials.gov will also be searched. Grey literature including dissertations, preprint repositories and conference proceedings will be searched. The key search terms include "stroke", "care integration", "continuity of care", "information exchange", "patient-centred care", "multidisciplinary care", "case management" and "low and middle-income countries", together with their alternative terms and synonyms, singular and plural forms and American and British spelling. Hand search of references of relevant studies will be carried out and experts in the field of stroke care coordination interventions will be contacted for their knowledge about any study missed by

**Data availability statement:** No datasets were generated or analysed during the current study. All relevant data from this study will be made available upon study completion.

**Funding:** The author(s) received no specific funding for this work.

**Competing interests:** The authors have declared that no competing interests exist.

**Abbreviations:** FIM, Functional Independence Measure; GRADE, Grading of Recommendations, Assessment, Development and Evaluation; ICTRP, International Clinical Trials Registry Platform; LMICs, Low and Middle-Income Countries; NIHSS, National Institute of Health Stroke Scale; PACTR, Pan African Clinical Trials Registry; PICO, Population Intervention Comparator Outcome; RCTs, Randomized Controlled Trials; RoB 2, Revised tool for Risk of Bias in randomized trial; ROBINS-E, Risk Of Bias in non-randomized Studies - of Exposures; ROBINS-I, Risk Of Bias in Non-randomized Studies - of Interventions; SoF, Summary of Findings; SSA, Sub-Saharan Africa; UGCESP, University of Ghana Centre for Evidence Synthesis and Policy; WHO, World Health Organization; WSO, World Stroke Organization

our searches. The studies will be collated in Rayyan and duplicates removed. Study selection will be done using a study selection flow chart developed from the pre-specified eligibility criteria defined by the PICOS elements (P − patient, I − intervention, C − comparator, O − outcomes and S −study). Quality in the included studies will be assessed for risk of bias using the Cochrane Risk of Bias tool (version 2) for Randomized Controlled Trials (RoB 2) and ROBINS-I for Non-randomized Studies of Intervention. Data will be extracted using a pre-tested data extraction form developed from Microsoft Excel. Study selection, data extraction and risk of bias assessment will be conducted independently by two reviewers (SKJO and ADA). Disagreements between the reviewers will be resolved through discussion. Risk ratio (RR) will be used as the effect measure for binary/dichotomous outcomes. For continuous outcomes mean difference (MD) with standard deviation (SD) or standardized mean difference (SMD) for outcomes measured on different instruments or scales will be used as effect measures for expressing effectiveness of care coordination interventions. Random-effects meta-analyses will be employed to pool studies, and all effects estimates will be reported with their 95% confidence interval (CI). Heterogeneity will be assessed using the $I^2$ statistic. The overall certainty of evidence will be assessed using the Grading of Recommendations Assessment, Development and Evaluation (GRADE).

## Expected outcomes

This review will attempt to identify the components of stroke-coordinated care interventions in low- and middle-income countries. It will assess whether these interventions improve clinical outcomes such as motor and cognitive functioning, mental wellbeing, mortality, and disability-adjusted life years (DALYs) of stroke survivors. The review will also explore how factors —such as stroke type, severity, and the nature of care coordination interventions (including the specific components) influence clinical, functional and psychosocial outcomes. The findings will help in determining the most effective care coordination interventions that should be adopted as care models to improve patient outcomes, whilst identifying evidence gaps for future research.

## Protocol registration and dissemination

This systematic review and meta-analysis protocol has been registered in PROSPERO [CRD42024587311]. The findings of the study will be shared with the relevant stakeholders and disseminated through scientific conferences and peer review publications.

## Introduction

Stroke remains a serious public health challenge, particularly in low and middle-income countries (LMICs). In 2019 alone, an estimated 6.5 million stroke-related

deaths were reported and around 101 million people were living with stroke, making stroke the third leading cause of death globally [1]. Low and Middle-Income Countries accounted for the bulk of these stroke related deaths (86%). The study also showed that LMICs were experiencing higher rate in stroke incidence (double fold) compared to 46% in high income countries between 1990–2019. The increasing burden of stroke, especially in LMICs, is a strong call for innovative care interventions that support care needs and improve patient outcomes [2].

Stroke, a neurological disorder resulting from an abrupt injury to cerebral blood vessels [3] is associated with acute and focal neurological deficits that could impact an individual's ability to perform physical (motor), and cognitive functions [4]. Major independent risk factors include hypertension, diabetes and lifestyle behaviours that increase the risk of cardiovascular conditions and diabetes mellitus [5,6]. The presentation of stroke includes unilateral weaknesses of limbs, speech disturbance (slurred speech), altered level of consciousness and headaches [3]. Non-contrast computed tomography (CT), and brain magnetic resonance imaging (MRI) scans remain gold standard diagnostic approaches for characterizing brain injury and making management decisions [7].

The management of stroke is complex and requires approaches that address varying clinical, physiological, nutritional and social needs of the patient delineated by four clinical stages: hyperacute, acute inpatient care, rehabilitation and longer-term stroke recovery [8]. The acute and hyperacute phases aim to stabilize the patient and limit brain damage. For ischemic stroke, this involves restoring blood flow with anticoagulants, thrombolytics like tPA (administered within three hours), or thrombectomy for larger clots. In haemorrhagic stroke, the priority shifts to stopping the bleed and reducing intracranial pressure with antihypertensives, platelet transfusions, or surgical interventions like hematoma evacuation [9–11]. As patients move beyond the acute phase, rehabilitation begins to play a key role, focusing on regaining function and independence [12]. Early interventions target mobility and cognitive function to prevent complications such as contractures or deep vein thrombosis [10]. Rehabilitation is adapted to each patient's deficits, incorporating therapies for physical, cognitive, and speech recovery. The long-term phase addresses ongoing recovery and reintegration. However, in LMICs, limited resources and gaps in service continuity often pose a serious challenge to comprehensive stroke care across all phases [8].

Stroke management in LMICs has improved over the years [8,13]. Increasingly, thrombolytic therapy for acute stroke and organized stroke care units in LMIC settings are being adopted, which have improved stroke outcomes such as decreased mortality and early discharge [3]. Despite these advances, stroke management in LMICs faces substantial challenges [2]. For example, very few facilities have organized stroke units and dedicated specialists for stroke management. These facilities also lack diagnostic equipment, stroke specialists, timely interventions such as tissue-type plasminogen activator (t-PA) and low rehabilitation access rates (as low as 40%) for stroke survivors in some countries [2]. In light of these challenges, LMICs require health system strengthening and effective care models to deliver optimal stroke services for improved outcomes. Care coordination emerges as effective model to enhance stroke care in these settings. This model of care is crucial as stroke survivors usually need to navigate multiple providers and services, especially during the post-emergency phase of stroke. In limited-resource settings, the seamlessness of this navigation may be compromised by disorganized or fragmented care, adversely affecting recovery outcomes. [14]. Considering this, global bodies including the WHO and World Stroke Organization (WSO) have provided recommendations for the implementation of coordinated care for delivering coherent and interconnected care that align with patient needs, values and preferences while attempting to improve outcomes [2,15,16]. Coordinated care also facilitates an environment of care continuity ensuring the needed long-term care support for stroke survivors [16]. Care coordination is a deliberate organization of patient care activities between two or more participants (typically between patients, caregivers/family and multiple providers) to facilitate appropriate delivery of services [17,18]. The organization of patient care activities involves information exchange between multiple providers and marshalling of care professionals and resources required for patient-centred care.

The concept of care coordination has evolved significantly since the 1970s, initially focusing on integrating healthcare tasks within organizations to improve efficiency. Early definitions emphasized aligning processes for task management

[19]. However, by the 1980s, as chronic disease management became more prominent, care coordination shifted toward ensuring continuity of care across different settings, highlighting the importance of information exchange between providers [20]. The 1990s saw a greater emphasis on communication between primary and specialist care providers, along with the introduction of case management models that integrated healthcare with social services, especially in mental health and chronic disease care [21,22]. This expanded the scope of care coordination beyond just healthcare providers, necessitating collaboration with community resources. By the late 1990s and early 2000s, system-wide coordination became critical due to the increasing complexity of healthcare systems, fragmentation of services and the need for coordinated efforts across organizational boundaries to ensure continuous care was stressed [23]. In the 2000s, multidisciplinary and interdisciplinary approaches became the norm, with team-based care models emphasizing collaboration, clear role understanding, and structured processes like the use of care coordinators. These developments reflected a shift from organizational efficiency toward patient-centred, comprehensive care, driven by the complexity of managing chronic diseases and the need for seamless, integrated care across multiple providers [17].

### How the intervention might work

Stroke care involves different health professionals such as neurologists, physiotherapists, speech therapists, primary care physicians and nurses etc. Care coordinated intervention aims to facilitate collaboration among these professionals and ensure aligned efforts (creation and implementation of a comprehensive care plan) that support patient recovery [24]. Care coordination intervention may also synchronize independent care activities such as medication management, physical rehabilitation and management of comorbidities [17]. Another component of care coordination is continuous exchange of information among health providers, patents, and caregivers. This ensures that all parties are informed about the status of the patient, treatment progress and evolving needs of the patient. The ultimate goal of care coordination is to integrate the services of different providers and provide seamless and patient-centred care.

The elements of care coordination (MDT collaboration, synchronization of independent care activities, timely information exchange, and integration of care across the continuum) ensure that planned care meets the complex needs of stroke survivors. The intervention also minimizes gaps in care that may lead to deterioration in physical and cognitive functioning of stroke survivors [25]. The integrated approach may reduce the likelihood of complications and recurrent strokes, as patients are provided continuous monitoring for adjustments for care plan based on evolving needs. Reduction of miscommunication due to care coordination interventions enhances clarity among multiple providers and reduces errors (e.g., medication errors) that may contribute to adverse outcomes and readmissions [26,27].

The benefit of stroke care coordination in LMICs settings is inconclusive. First, care coordination notably has variations in their structures, delivery and outcomes across different settings [28,29]. Earlier systematic reviews (five) attempted to synthesize evidence on the effectiveness of care coordination interventions on patient outcomes, but three of these systematic reviews were conducted more than a decade ago and are outdated [30–32]. Of the only two available recent systematic reviews, one review focused on case management, a co-intervention of care coordination [33] and the other looked at effectiveness of care coordination among patient groups other than stroke survivors [34]. Evidently, the most prominent gap is that of limited evidence on care coordination intervention effectiveness in LMICs.

The current review will aim to provide a robust synthesis on the effectiveness of care coordination intervention of stroke survivors in LMIC settings. The sources of variation in care coordination outcome will also be explored. The review is crucial for informing clinical practice and guidelines of action such as those by the WSO, particularly in settings where resources for stroke care are limited. The synthesis aligns with SDG 3 and 10 which emphasize good health and well-being and reduced inequalities. The evidence generated could inform the design of context-sensitive stroke coordinated care models in resource limited settings for stroke survivors that will ensure they receive timely, effective and continuous care, ultimately leading to optimal health outcomes [16].

The review aims to address the following questions: 1) What are the components of stroke care coordination interventions delivered in LMICs? 2) Does care coordination improve clinical outcomes of stroke survivors in similar facilities and settings compared to other care models? 3) What is the effect of care coordination interventions on quality of life of stroke patients? 4) Do clinical stroke types (ischaemic and haemorrhagic), subtypes (intracerebral and subarachnoid haemorrhage, small vessel blockade etc.) and severity (mild, moderate and severe/critical) influence patient outcomes? The specific objectives are to: 1) Identify the components of stroke care coordination interventions delivered in low and middle-income countries, 2) Evaluate effectiveness of care coordination on clinical outcomes of stroke survivors in similar facilities and settings compared to other care models, 3) Assess the effectiveness of care coordination interventions on the quality of life of stroke, and 4) investigate impact of clinical stroke types and subtypes (ischaemic and haemorrhagic), subtypes (intracerebral and, subarachnoid haemorrhage, small vessel blockade etc.) and severity (mild, moderate and severe/critical) on patient outcomes.

## Methods

The systematic review and meta-analysis were conducted in accordance with Cochrane methods for systematic reviews of interventions [35] and earlier published systematic review [36–38]. The reporting of the protocol was guided by the Preferred Reporting Items for Systematic Review and Meta-Analysis extension for protocols (PRISMA-P) [39] (S1 Table). The flow of studies from the searches through the selection process will be reported using the PRISMA Flow Diagram [40] (S1 Fig).

### Patient and public involvement

Individual consultation of stroke survivors and the public was not carried out. However, relevant literature on patient expectations of neurorehabilitation and multidisciplinary stroke interventions were reviewed to support the selection of outcomes considered to be important to patients and caregivers. The interventions being assessed in this review include telerehabilitation, multidisciplinary rehabilitation, early supported discharge and mindfulness program [41–45]. Improvement in physical and cognitive functioning is a prime expectation of stroke survivors' post-acute management [41–44]. Stroke is an abrupt condition that leaves patients emotionally distressed [41] and as such, improvement in mental well-being is a major expectation of stroke survivors receiving varying interventions [41,45]. Other expectations included better knowledge and improved self-care skills, social well-being and motivation [41–43]. In this systematic review, outcomes relevant to the patient and their caregivers, and reflecting the patient's values, beliefs and expectations have been discussed and included in the context of patient and public involvement.

### Criteria for considering studies for this review

**Type of studies.** Randomized controlled trials (RCTs), quasi-RCTs, and other prospective studies (e.g., cohort studies) reporting on any care coordination interventions implemented in post-emergency stroke care in LMICs will be considered for inclusion. Reviews on these interventions will not be included; however, their reference lists will be screened for potential eligible primary studies missed in our searches. Expert opinions, commentaries, newsletters, case series, and case studies will be excluded.

**Population.** Adult stroke survivors (all stroke types and sub-types) who received care coordination in any low and middle-income country will be eligible for inclusion. Stroke survivors in emergency states will not be eligible for inclusion as care coordination is typically not the primary focus during this phase, and the outcomes of interest in this review are more relevant to post-emergency care.

**Intervention***: In the context of this systematic review, care coordination is defined as a deliberate organization of care activities (i.e., structured, patient-centred tasks and interventions performed to address ongoing medical, functional, psychological and social needs of stroke survivors) involving the patient family members and multiple healthcare providers including primary physicians, neurologists, physiotherapists, nurses, social workers, speech therapists etc.. Care

coordination may be executed differently in different contexts. As such, care coordination interventions will be considered for inclusion if they have the core components proposed by McDonald et al. [32]: 1) there should be multiple participants (patients, families, health and care providers) involvement in care activities, 2) an established mechanism of information exchange (such as interprofessional meetings, exchange of information via electronic health systems etc.), and 3) clarification of participants roles and responsibilities (e.g., orientation, outlining of responsibilities by a coordinator, using checklists to assign specific tasks etc.) and available resources. In addition to these core components, care coordination may have at least one of the supporting features from the three categories described by Karam et al. [46]: 1) *targeting patients and families* to include assessing patient needs, developing care plans, providing direct care, and monitoring responses to care, 2) *linking patients with services and the multidisciplinary team (MDT)* to encompass partnering with community resources, fostering collaboration within and across MDTs, and facilitating care transitions, and 3) *targeting the MDT* activities that should focus on clarifying roles, negotiating responsibilities, establishing shared responsibilities, and exercising leadership. Care coordination interventions may have coordinators, usually a nurse or social worker trained for the role. However, this role may not be explicit in all care coordination interventions.

**Comparator.** The comparators will be usual or standard care/fragmented care lacking the core components of care coordination.

**Outcomes.** Outcomes of interest will be those that are relevant to the patients and their caregivers and should reflect on patient expectations of care models and indicators of effectiveness of care coordination in stroke management. This review focuses on patient clinical outcomes, quality of life (QoL), self-efficacy and experience with care and service use.
*Primary outcomes:* The primary outcomes are those directly related to patients clinical and functional status. They include mortality, functional recovery and independence (motor recovery, activities of daily living performance, and balance), cognitive recovery (executive function and speech and language outcomes) and cardiovascular risk management (blood pressure control, blood glucose control and lipid management). These outcomes have been defined or explained below.

- **Mortality:** rates of stroke related deaths during the follow-up period of the study.

- **Readmission:** Rates of unplanned rehospitalization of stroke survivors.

- **Length of hospital stay:** the number of days survivors spend hospitalised during the acute and sub-acute phases of stroke.

- **Stroke severity:** improvement in the severity of stroke symptoms as assessed by validated scales.

- **Motor recovery and independence:** changes in upper and lower extremity motor recovery, gait and balance recovery and cognitive function improvement (executive functioning and speech and language) as measured by appropriate validated clinical tools. Improvement in capacity to perform ADL measured by validated scales.

- **Cardiovascular risk management**: changes in systolic/diastolic blood pressure, glycated haemoglobin ($HbA_1C$) and lipid profiles (total cholesterol, LDL, HDL, and triglycerides).

*Secondary outcomes:* Secondary outcomes are patient-centred and healthcare system effects of care coordination. In this review, secondary outcomes include QoL, mental health outcomes of patients and caregivers, health service utilisation, patient satisfaction with care and community reintegration. The outcomes have been defined and/or explained.

- **Quality of life**: Changes in overall quality of life or quality of life pertaining to a particular domain, using validated tools.

- **Mental health**: Incidence of depression and anxiety in the patients and caregivers.

- **Health service:** The number of scheduled or unscheduled outpatient visits to healthcare providers (e.g., rehabilitation, primary care) following stroke discharge, and unplanned or emergency hospital admissions within the study follow-up period, particularly for stroke-related complications.

- **Patient satisfaction:** Patient report on how well care aligns with their needs, expectations and experiences as evaluated by surveys or validated tools.

- **Community integration:** The differences in the extent at which patients are able to participate in meaningful social, occupational, and community activities as evaluated by validated tools.

  **Adverse events.** All adverse events reported in the included studies, especially from the RCTs, will be summarized in this systematic review. Any serious adverse events (SAEs)—those leading to death, hospitalization, disability, or significant harm—will be specifically collated and their impact. on the care coordination intervention evaluated where information is available.

  **Setting.** This systematic review will include studies conducted in LMICs (based on the World Bank classification of countries) [47]. Low and Middle-Income Countries have limited stroke services capacity and rely on effective care delivery models (including multidisciplinary team care, specialist-led care, physician led care, task sharing and shifting, and hub and spoke model of care) to ensure optimal and quality stroke care [2]. Health systems in LMICs are not entirely homogeneous. However, these health systems share cross-cutting challenges including lack of resources and institutional capacity and reliance on international actors (example financing by international donors) [48]. The limited rehabilitation facilities, fewer stroke specialists and barriers to follow-up care may directly impact stroke care [2]. This systematic review will focus on both community and healthcare settings that served as the study sites for stroke care coordination interventions. For healthcare settings, studies carried out in in-patient units, rehabilitation centres, long-term stroke facilities, and outpatient departments will be considered.

  **Search strategy.** Databases and non-database sources will be searched for published and unpublished studies. We will search PubMed, LILACS, Google Scholar, CINAHL via EBSCOhost, Scopus and Cochrane CENTRAL from 2000 (when multidisciplinary elements and care integration became formalized) to 31st May 2025, without language restriction. Trial registries including the WHO International Clinical Trials Registry Platform (ICTRP) which hosts multiple registries across the globe including the Pan African Trial Registry (PACTR) and Clinicaltrials.gov by the US government will also be searched. Grey literature including dissertations, preprint repositories and conference proceedings will be searched. The search terms include "stroke", "care integration", "continuity of care", "information exchange", "patient-centred care", "multidisciplinary care", "case management", together with their alternative terms and synonyms, singular and plural forms, American and British spelling and "low and middle-income countries". The search terms and their alternatives will be combined using Boolean operators to form the search strategy developed for PubMed (Table 1), which will be adapted for other databases. Reference list of relevant articles will be screened for additional studies and experts in the field of stroke care coordination will be contacted, where necessary, for their knowledge about any study missed by our searches.

## Managing the search results and selecting studies

The search results will be managed using EndNote where duplicate records will be removed. The deduplicated records will then be exported to Rayyan (a web-based platform for organizing, managing and collaborating systematic reviews) for screening and study selection using the study selection flow chart (Fig 1). The titles and abstracts of studies will be initially screened by two independent reviewers and the potentially relevant studies will be considered for full-text screening for eligibility. Any disagreements or discrepancies between reviewers will be discussed and resolved among the reviewers.

## Data extraction and management

Data from included studies will be extracted independently by two reviewers using an adapted Cochrane data collection form [49] (S2 Table). The adapted form will be pretested with three of the included studies and necessary modification will be made prior to full-scale extraction. The form captures study ID (author name and year), design, unit of allocation, study duration, population description, setting, eligibility criteria, method of recruitment, sample size, baseline characteristics

**Table 1. Search strategy for PubMed.**

| Concept | Query | Results |
|---|---|---|
| #1 Stroke | ("stroke"[tw] OR "cerebrovascular accident"[tw] OR CVA[tw] OR "transient ischemic attack"[tw] OR TIA[tw] OR stroke[mesh]) | |
| #2 Care Coordination | ("Integration care"[Title/Abstract:~3] OR "integrat* care" OR "care integrat*" OR "Integration service"[Title/Abstract:~3] OR "Integration services"[Title/Abstract:~3] OR "integrat* service*" OR "care integrat*" OR "service integrat*" OR "Delivery of Health Care, Integrated"[mesh] OR "care continuity" OR "continuity care"[Title/Abstract:~3] OR "Continuity of Patient Care"[Mesh] OR "Health Information Interoperability"[Mesh] OR "Health Information Exchange"[Mesh] OR "exchange information"[Title/Abstract:~3] OR Patient-centered OR "patient centred" OR "patient centered" OR patient-centred OR multidisciplinary OR "collaborative care" OR team-based OR "team based" OR "case management"[mesh] OR "Care coordinat*" OR "care co-ordinat*" OR "coordinated care" OR "co-ordinated care" OR "coordinating care" OR "co-ordinating care" OR "coordination care"[Title/Abstract:~3] OR "co-ordination care"[Title/Abstract:~3] OR "coordinating care"[Title/Abstract:~3] OR "co-ordinating care"[Title/Abstract:~3] OR "coordinator care"[Title/Abstract:~3] OR "care coordinat*" OR "patient navigat*") | |
| #3 | #1 AND #2 | |
| #4 LIMCs | ("Developing Countries"[Mesh] OR "Africa South of the Sahara"[Mesh] OR "Africa"[Mesh] OR "Latin America"[Mesh] OR "Asia, Southern"[Mesh] OR "Sub-Saharan Africa" OR "South Asia" OR "Southeast Asia" OR "Latin America" OR "Caribbean Region" OR "West Africa" OR "Central Africa" OR "East Africa" OR "Southern Africa" OR "Pacific Islands" OR Afghanistan OR Albania OR Algeria OR Angola OR Argentina OR Armenia OR Azerbaijan OR Bangladesh OR Barbados OR Benin OR Belarus OR Belize OR Bhutan OR Bolivia OR Bosnia OR Herzegovina OR Botswana OR Brazil OR Burkina Faso OR Burundi OR Cambodia OR Cameroon OR "Cape Verde" OR "Central African Republic" OR Chad OR Chile OR China OR Colombia OR Comoros OR Congo OR "Cote d'Ivoire" OR Cuba OR Djibouti OR "Dominican Republic" OR Ecuador OR Egypt OR El Salvador OR Eritrea OR Ethiopia OR Fiji OR Gabon OR Gambia OR Georgia OR Ghana OR Guatemala OR Guinea OR Guyana OR Haiti OR Honduras OR India OR Indonesia OR Iran OR Iraq OR Jamaica OR Jordan OR Kazakhstan OR Kenya OR Kiribati OR Korea OR Kosovo OR Kyrgyzstan OR Laos OR Lebanon OR Lesotho OR Liberia OR Libya OR Madagascar OR Malawi OR Malaysia OR Maldives OR Mali OR Marshall Islands OR Mauritania OR Mauritius OR Mexico OR Micronesia OR Moldova OR Mongolia OR Montenegro OR Morocco OR Mozambique OR Myanmar OR Namibia OR Nepal OR Nicaragua OR Niger OR Nigeria OR Pakistan OR Palau OR Panama OR Papua New Guinea OR Paraguay OR Peru OR Philippines OR Rwanda OR Samoa OR "Sao Tome and Principe" OR Senegal OR Serbia OR Seychelles OR "Sierra Leone" OR "Solomon Islands" OR Somalia OR "South Africa" OR "Sri Lanka" OR Sudan OR Suriname OR Swaziland OR Syria OR Tajikistan OR Tanzania OR Thailand OR Timor-Leste OR Togo OR Tonga OR Tunisia OR Turkey OR Turkmenistan OR Tuvalu OR Uganda OR Ukraine OR Uruguay OR Uzbekistan OR Vanuatu OR Venezuela OR Vietnam OR Yemen OR Zambia OR Zimbabwe) | |
| #5 | #3 AND #4 | |

(age, sex, race, stroke type and severity), intervention (components and activities), theoretical basis, duration of treatment, timing, and providers involved. Information on outcomes definitions and criteria, person measuring outcomes, tools for measuring, results of outcomes and time points of measurement. Extracted data will be verified using techniques including range check, missing data screening and confirmation, consistency checks and other validation approaches. Inconsistencies between independent extractions will be cross-checked by verifying the data from the full-text study and any disagreements will be resolved through discussion. Authors of studies with pertinent information missing (e.g., details needed for calculating effect estimates) will be contacted to see if they can provide the missing data. If there is no feedback from corresponding authors, missing data will not be computed.

## Risk of bias assessment

The revised tool for Risk of Bias in randomized trials (RoB 2) [50] (S3 Table) and Risk of Bias in Non-randomized studies of intervention (ROBINS-I) tools [51] (S4 Table) will be used to assess risk of bias in each of the included studies. The RoB 2 is structured in five domains through which bias might be introduced into results of the included randomized trials. The domains are 1) risk of bias arising from randomization process, 2) risk of bias due to deviation from intended intervention, 3) risk of bias due to missing outcome data, 4) risk of bias in measure of outcome and 5) risk of bias in selecting the reported results. Each domain has a series of signaling questions to elicit information for assessment of risk of bias. For

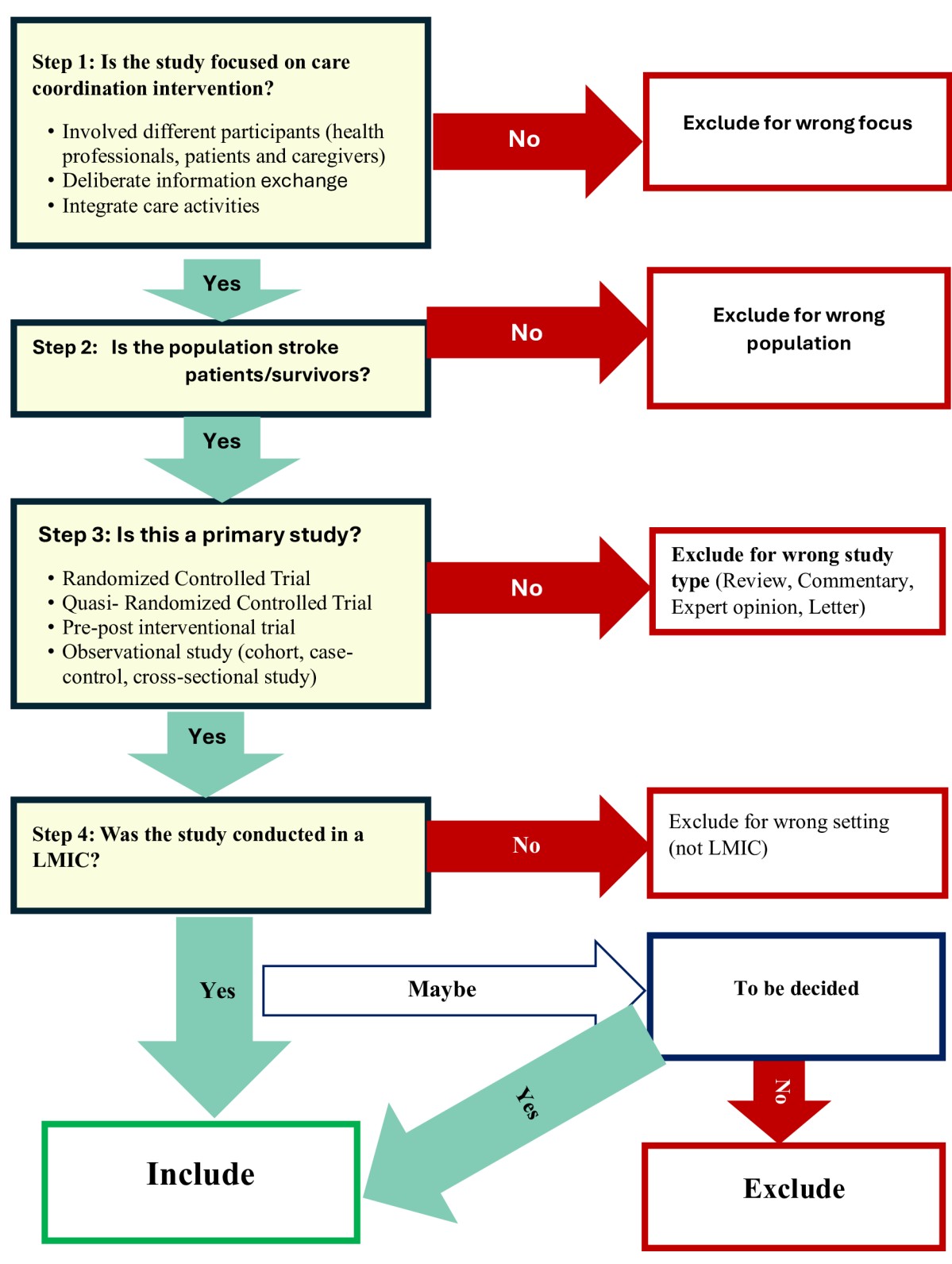

**Fig 1. Study selection flowchart.**

each domain, a judgment of 'low risk' of bias, 'high risk' of bias or 'some concerns' will be made. The judgement for each domain will be used to inform the overall risk of bias in an included study. ROBINS-I has seven domains that focus on biases due to 1) confounding, 2) selection of participants into the study, 3) classification of interventions 4) deviations from intended interventions, 5) missing data, 6) measurement of outcomes and 7) selection of the reported results. The same process as RoB 2 will be followed to assign the overall risk of bias in the included studies. Risk of bias assessment will be carried out by two independent reviewers and disagreements will be either resolved through discussion.

## Plan of data analysis

For continuous outcomes, mean difference and standard deviation (SD) will be computed from the included studies if they used the same scale for measuring an outcome, otherwise standardized mean difference (SMD) will be used when an outcome is measured on different scales or instruments. For binary or dichotomous outcomes, we will compute risk ratio (RR) as the effect measure for expressing effectiveness of care coordination interventions. Random-effects model will be employed in the meta-analysis to pool studies to estimate pooled effects of care coordination interventions. Intention-to-treat (ITT) analysis will be conducted but when this is not suitable based on the magnitude of missing data, complete case or per-protocol analysis based on the number of patients for whom an outcome was recorded will be conducted. All pooled effect estimates will be reported with their 95% confidence intervals (CIs). The pooled effect estimates will be displayed using forest plots. In situations where a meta-analysis is not appropriate, a narrative synthesis will be used to summarize the evidence across studies.

## Heterogeneity and subgroup analysis

Heterogeneity may arise from variation in the study setting, characteristics of participants, design, intervention components or outcomes to be assessed across studies. The proportion of the variation that can be explained by real differences in the studies will be assessed using $I^2$ statistic. A sub-group analysis will be conducted to explore the potential impact of heterogeneity on the pooled effect estimates. The factors to be considered for sub-group analysis include duration of intervention, study design, stroke severity, components of care coordination, setting, and designated care coordinator.

## Sensitivity analysis

Sensitivity analysis will be performed to evaluate the robustness of the pooled effect estimates on the quality domains and outliers. Outlier studies will be identified based on extreme effect sizes or other factors such as methodological quality, or sample size that could disproportionately influence the overall findings.

## Managing missing data

Missing data will not be imputed, instead, we will contact the primary study authors for the missing data or information. Where missing data cannot be obtained, intention-to-treat (ITT) analysis will be conducted if the data permit.

## Grading the evidence

The quality of evidence reflects the extent to which the reviewers are confident that a pooled estimate is accurate. The Grading of Recommendations, Assessment, Development and Evaluation (GRADE) methodology for judging the quality of evidence on outcomes from care coordination interventions. A summary of findings (SoF) table will be provided to provide a summary of findings (list of outcomes, number of studies and total sample size, absolute/relative effects, quality of evidence and explanatory notes on judgement) for each of the included outcomes. The rating of quality of evidence on each outcome will be based on risk of bias, inconsistence of results, indirectness of evidence, imprecision and publication bias. Grading ratings could be high, moderate, low or very low [52].

## Ethics statement

This study does not require ethical approval as it involves the synthesis of publicly available data from already completed studies and does not involve primary data collection or direct contacts with human subjects.

## Discussion

Stroke survivors have to navigate multiple providers and services as part of the care and recovery process [4,14]. Care coordination is essential in ensuring interconnected care that aligns with patient needs and ultimately supports optimal recovery outcomes. However, the effectiveness of care coordination interventions and their character in low- and middle-income settings particularly for stroke has not been comprehensively explored [33,34]. The current study will characterize stroke care coordination and its effectiveness in low resource settings.

The findings of the study are expected to offer meaningful implications for both clinical practice and stroke related health policy in LMICs. Specifically, they could guide context-specific clinical guidelines and evidence-based protocols on implementing care models for stroke management. Also, these findings could inform clinical decisions to adopt care coordination models as the review will highlight the nuances (e.g., patient groups, components, and personnel) of care coordination and their influence on effectiveness. For policymakers, the evidence could support the integration of coordinated care models into stroke national strategies. This may involve allocation of funding to support care coordination support infrastructure, such as electronic health systems.

## Study limitations

We anticipate some challenges in this review. One primary challenge is the anticipated variation in study designs and characteristics of care coordination interventions across LMICs. This variability may complicate the analysis of results or even lead to misclassifications during data extraction processes. The quality of studies carried out in LMICs may vary, leading to potential biases in pooled results. We will address these limitations through a thorough risk of bias assessment and sensitivity analysis. Additionally, crucial long-term outcomes such as mortality over time may be scarce due to limited availability of long-term follow up data essentially preventing us from understanding the durability of care coordination impact in stroke care management.

## Supporting information

**S1 Fig. PRISMA flow diagram.**
(PDF)

**S1 Table. PRISMA-P GUIDELINES.**
(PDF)

**S2 Table. Cochrane data extraction form.**
(PDF)

**S3 Table. Cochrane risk of bias tool for RCTs.**
(PDF)

**S4 Table. Cochrane risk of bias tool for non-randomized studies of intervention (ROBINS-I).**
(PDF)

## Acknowledgments

This systematic review protocol was part of the joint capacity building initiative by the Centre for Evidence Synthesis and Policy, University of Ghana (CESP-UG) and Africa Communities of Evidence Synthesis and Translation (ACEST) that train in evidence Synthesis and Translation across Low and Middle-Income Countries, particularly Sub-Saharan Africa.

Stephanopoulos Kofi Junior Osei is a clinical trialist who is being mentored in Evidence Synthesis and Translation by Prof. Anthony Danso-Appiah (Director, Centre for Evidence Synthesis and Policy, University of Ghana).

## Author contributions

**Conceptualization:** Stephanopoulos Kofi Junior Osei, Anthony Danso-Appiah.

**Investigation:** Stephanopoulos Kofi Junior Osei, Anthony Danso-Appiah.

**Methodology:** Anthony Danso-Appiah.

**Project administration:** Stephanopoulos Kofi Junior Osei, Anthony Danso-Appiah.

**Resources:** Anthony Danso-Appiah.

**Supervision:** Anthony Danso-Appiah.

**Validation:** Stephanopoulos Kofi Junior Osei, Anthony Danso-Appiah.

**Writing – original draft:** Stephanopoulos Kofi Junior Osei, Anthony Danso-Appiah.

**Writing – review & editing:** Stephanopoulos Kofi Junior Osei, Anthony Danso-Appiah.

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
