## [Decision Letter · Decision Letter 0]

12 Jan 2025

PONE-D-24-49441Effectiveness of Coordinated Care Interventions Delivered to Stroke Survivors in Low and Middle-Income Countries: Systematic Review and Meta-Analysis ProtocolPLOS ONE

Dear Dr. Danso-Appiah,

Thank you for submitting your manuscript to PLOS ONE. After careful consideration, we feel that it has merit but does not fully meet PLOS ONE’s publication criteria as it currently stands. Therefore, we invite you to submit a revised version of the manuscript that addresses the points raised during the review process.

 In addition to the reviewer comments below, I have also appended a few comments from the editor. My general impression of this protocol is that there is an overuse of vague language and methodological descriptions. Because this is a protocol paper, it is very important for your methods to be concretely defined and set in stone. Please review your protocol carefully to check for this during your revision. Please submit your revised manuscript by Feb 26 2025 11:59PM. If you will need more time than this to complete your revisions, please reply to this message or contact the journal office at plosone@plos.org . Please include the following items when submitting your revised manuscript:

We look forward to receiving your revised manuscript.

Kind regards,

Jiawen Deng

Academic Editor

PLOS ONE

Journal Requirements:

Additional Editor Comments:

1. I question the use of Google Scholar as a literature source. As you may know, most entries on the Google Scholar database are unmoderated and may not be legitimate research articles. It is also difficult to develop systematic search strategies for Google Scholar compared to dedicated academic database providers, such as OVID or EBSCOhost.

2. Please concretely define whether you will compute OR or RR as the effect measure for dichotomous outcomes.

3. Line 418, please remove capitalization on "Forest plots".

4. Please do not use vague language such as "The synthesis may be stratified based on patient populations, components of interventions and study designs." Given that this is a prospective protocol, you should clearly define any subgroup or sensitivity analysis that you are going to conduct.

5. Note that you should not be using statistical measures of heterogeneity as the deciding factor for whether a random- or fixed-effects approach is used. This decision should be purely made based on characteristics of the included patients and studies -- in most cases, this will be a random-effects model unless you have a clear justification for using fixed-effects (e.g., the patient population are very similar, such as pooling studies from the same institution). Please concretely define your approach as a random-effects approach.

6. What is "appropriate imputation methods"? Clearly define them and any sensitivity analyses that will be used to assess impact of imputation, and cite any methodological paper, if needed.

Reviewers' comments:

Reviewer's Responses to Questions

**Comments to the Author**

1. Does the manuscript provide a valid rationale for the proposed study, with clearly identified and justified research questions?

Reviewer #1: Yes

Reviewer #2: Yes

2. Is the protocol technically sound and planned in a manner that will lead to a meaningful outcome and allow testing the stated hypotheses?

Reviewer #1: Yes

Reviewer #2: Yes

3. Is the methodology feasible and described in sufficient detail to allow the work to be replicable?

Reviewer #1: Yes

Reviewer #2: No

4. Have the authors described where all data underlying the findings will be made available when the study is complete?

Reviewer #1: Yes

Reviewer #2: Yes

5. Is the manuscript presented in an intelligible fashion and written in standard English?

Reviewer #1: Yes

Reviewer #2: Yes

6. Review Comments to the Author

You may also provide optional suggestions and comments to authors that they might find helpful in planning their study.

Reviewer #1: The manuscript describes a study protocol that aims to assess the effectiveness of coordinated care interventions delivered to stroke survivors in low- and middle-income countries through a systematic review and meta-analysis.

The protocol is very well written and systematically designed in accordance with the latest guidelines. Please find below some minor comments:

1. If you hypothesize having outcomes measured on different scales, I suggest including the SMD, which converts all these measures into standard units for the analysis of continuous outcomes.

2. I recommend aligning the methods described in the abstract with those detailed in the full text (e.g., the electronic databases listed are not entirely consistent, and key searches in LMIC-related field are not mentioned).

3. I suggest ensuring consistency between this protocol and its registration on PROSPERO. This could be achieved by making minor updates to the PROSPERO registration, such as finalizing the electronic search strategy, including the GRADE approach, etc. Consistency across all documents is essential.

Reviewer #2: General Comments:

This manuscript presents a promising protocol for a systematic review and meta-analysis to assess the effectiveness of coordinated care interventions for stroke survivors in low- and middle-income countries (LMICs). The importance of this topic is evident given the increasing burden of stroke and the complex care needs of survivors in LMICs. The review has the potential to make a significant contribution to the literature, particularly in identifying effective care models for stroke survivors in resource-limited settings. However, several aspects of the manuscript require clarification, refinement, and additional detail. Below are my detailed comments and suggestions for revision:

Abstract:

The abstract provides a succinct overview of the study, but it could benefit from more clarity in describing the specific aims of the review. For example, it would be helpful to highlight explicitly that the review aims to identify the components of care coordination interventions and their impact on stroke outcomes.

The expected outcomes section could also be slightly reworded to clarify the role of stroke severity, type, and the nature of the interventions in influencing outcomes.

Suggested Revision:

Add more specific details in the aims section, such as "to assess the impact of care coordination interventions on clinical outcomes such as motor recovery, cognitive function, and mental health, while also exploring factors like stroke type and severity."

Background

Expand the background to include more specific details about the healthcare challenges in LMICs, such as a lack of stroke specialists, rehabilitation facilities, and access to follow-up care, which necessitate the need for effective care coordination.

METHODS

Suggested Revision:

Remove redundant mentions of databases like "ProQuest" and "Google Scholar."

Consider including preprint databases in the search strategy to capture emerging studies.

Inclusion/Exclusion Criteria:

While the inclusion criteria are well outlined, it is unclear how the study will define "care coordination" in the context of stroke management. The term “care coordination” is widely used but can have various meanings across studies. A clear, operational definition would help in the screening process.

Suggested Revision:

Define "care coordination" clearly, specifying which elements (e.g., case management, multidisciplinary care) must be present for a study to be included.

Risk of Bias:

The manuscript mentions that two reviewers will conduct risk of bias assessments but does not specify which validated tools will be used. It would be helpful to state explicitly which risk of bias tool will be used (e.g., Cochrane Risk of Bias tool for randomized trials).

Suggested Revision:

Specify the risk of bias assessment tools that will be used, such as the Cochrane Risk of Bias tool for randomized controlled trials (RCTs).

Suggested Revision:

OUTCOMES

Clarify if all the mentioned tests for motor recovery and QoL will be included or if a subset will be prioritized based on relevance to the intervention or study design.

Suggested Revision:

Provide more specific definitions for secondary outcomes like "service use" and "recurrent stroke" to clarify what will be included in the review.

Statistical Methods:

The statistical methods section mentions using a random-effects model when heterogeneity is significant but does not specify what thresholds for heterogeneity (e.g., I² statistic) would trigger this model. It would also be useful to explain how potential publication bias will be assessed.

Suggested Revision:

Specify the thresholds for I² that would justify the use of a random-effects model. Also, describe how publication bias will be assessed (e.g., funnel plots, Egger’s test).

The manuscript states that ethical approval is not required because the study involves secondary data. While this is understandable, it would be important to mention any potential conflicts of interest related to the inclusion of data from international donors or stakeholders who may influence stroke care policies.

Suggested Revision:

Include a brief statement regarding potential conflicts of interest, particularly related to the involvement of international funding or healthcare organizations.

Conclusion and Implications:

The expected outcomes are promising, but the manuscript could benefit from a clearer discussion on how the findings will specifically influence policy and practice in LMICs. For example, what actionable recommendations might emerge for stroke care teams or health policymakers?

Suggested Revision:

Expand the conclusion to include potential actionable recommendations for stroke care providers and policymakers in LMICs based on the expected findings.

Suggested Revision:

Conduct a thorough review for grammar and clarity, ensuring consistency in writing style and tone throughout the manuscript.

7. PLOS authors have the option to publish the peer review history of their article (what does this mean? ). If published, this will include your full peer review and any attached files.

**Do you want your identity to be public for this peer review?** For information about this choice, including consent withdrawal, please see our Privacy Policy .

Reviewer #1: No

Reviewer #2: No

---

## [Author Response · Author response to Decision Letter 1]

26 Mar 2025

Response to Reviewers’ Comments

Reviewer #1

1. If you hypothesize having outcomes measured on different scales, I suggest including the SMD, which converts all these measures into standard units for the analysis of continuous outcomes.

Standardized mean difference (SMD) has been stated in the main methods as the effect measure for outcomes measured on different instruments or scales (Lines 491-492). We have added this statement to the abstract (Line 66).

2. I recommend aligning the methods described in the abstract with those detailed in the full text (e.g., the electronic databases listed are not entirely consistent, and key searches in LMIC-related field are not mentioned).

Thank you, the methods described in the abstract have been aligned with those detailed in the full text to match.

3. I suggest ensuring consistency between this protocol and its registration on PROSPERO. This could be achieved by making minor updates to the PROSPERO registration, such as finalizing the electronic search strategy, including the GRADE approach, etc. Consistency across all documents is essential.

The registered protocol has been updated to match the manuscript.

Reviewer #2

1. The abstract provides a succinct overview of the study, but it could benefit from more clarity in describing the specific aims of the review. For example, it would be helpful to highlight explicitly that the review aims to identify the components of care coordination interventions and their impact on stroke outcomes.

The abstract has been revised to highlight the specific aims of the review. An objective “To identify the components of care coordination interventions” has been added (Line 32-36).

2. The expected outcomes section could also be slightly reworded to clarify the role of stroke severity, type, and the nature of the interventions in influencing outcomes.

The expected outcomes section of the abstract has been reworded to clarify the role of stroke severity, type, and the nature of the intervention in influencing stroke outcomes (line 36-37).

3. Add more specific details in the aims section, such as "to assess the impact of care coordination interventions on clinical outcomes such as motor recovery, cognitive function, and mental health, while also exploring factors like stroke type and severity."

Specific details have been added to the aims section of the abstract (Lines 36-37).

4. Background

Expand the background to include more specific details about the healthcare challenges in LMICs, such as a lack of stroke specialists, rehabilitation facilities, and access to follow-up care, which necessitate the need for effective care coordination

The background has been revised to highlight challenges in stroke management in LMICs which necessitate effective care coordination (line 141-155).

5. Remove redundant mentions of databases like "ProQuest" and "Google Scholar."

ProQuest has been removed but not Google Scholar, please see response to Editor’s comment #1.

6. Consider including preprint databases in the search strategy to capture emerging studies.

Thank you, preprint databases have been added as sources for retrieving studies (Line 45 and 431).

7. Inclusion/Exclusion Criteria:

While the inclusion criteria are well outlined, it is unclear how the study will define "care coordination" in the context of stroke management. The term “care coordination” is widely used but can have various meanings across studies. A clear, operational definition would help in the screening process.

Suggested Revision:

Define "care coordination" clearly, specifying which elements (e.g., case management, multidisciplinary care) must be present for a study to be included.

Thank you, we have defined care coordination in the context of stroke management and described the operational elements such as multiple participant involvement, established communication and role clarification. Although care coordination may be implemented with other care models such as multidisciplinary care and case management, the core elements specified in the main text must be present for a study to be considered eligible for inclusion (Line 278-297).

8. Risk of Bias:

The manuscript mentions that two reviewers will conduct risk of bias assessments but does not specify which validated tools will be used. It would be helpful to state explicitly which risk of bias tool will be used (e.g., Cochrane Risk of Bias tool for randomized trials).

Suggested Revision:

Specify the risk of bias assessment tools that will be used, such as the Cochrane Risk of Bias tool for randomized controlled trials (RCTs).

Thank you, the tools to be used for assessing risk of bias have been specified as RoB 2 for Randomized Controlled Trials (RoB 2) and ROBINS-I for Non-randomized Studies of Intervention (Line 59-61).

9. Clarify if all the mentioned tests for motor recovery and QoL will be included or if a subset will be prioritized based on relevance to the intervention or study design.

Suggested Revision:

Provide more specific definitions for secondary outcomes like "service use" and "recurrent stroke" to clarify what will be included in the review.

We have provided further information on secondary outcomes (service utilization and recurrent stroke).

10. Statistical Methods:

The statistical methods section mentions using a random-effects model when heterogeneity is significant but does not specify what thresholds for heterogeneity (e.g., I² statistic) would trigger this model. It would also be useful to explain how potential publication bias will be assessed.

Suggested Revision:

Specify the thresholds for I² that would justify the use of a random-effects model. Also, describe how publication bias will be assessed (e.g., funnel plots, Egger’s test).

We have revised our statement and random-effects meta-analysis will be conducted. I2 statistic will not be used as the basis for deciding which model (random-effects or fixed-effect model) will be used to pool studies. We have a very comprehensive search strategy and believe it is enough to prevent or minimize publication bias. We do not think formal analyses (e.g., funnel plots, Egger’s test) are necessary.

11. The manuscript states that ethical approval is not required because the study involves secondary data. While this is understandable, it would be important to mention any potential conflicts of interest related to the inclusion of data from international donors or stakeholders who may influence stroke care policies.

Suggested Revision:

Include a brief statement regarding potential conflicts of interest, particularly related to the involvement of international funding or healthcare organizations.

We appreciate the concerns raised about potential conflicts of interest, particularly related to the involvement of international funding or healthcare organizations. The potential bias such studies may present will be addressed in the risk of bias assessment.

12. Conclusion and Implications:

The expected outcomes are promising, but the manuscript could benefit from a clearer discussion on how the findings will specifically influence policy and practice in LMICs. For example, what actionable recommendations might emerge for stroke care teams or health policymakers?

Suggested Revision:

Expand the conclusion to include potential actionable recommendations for stroke care providers and policymakers in LMICs based on the expected findings.

Thank you, whilst the main review has not been conducted and we are reluctant to pre-empt the findings and implications, we have highlighted potential recommendations for clinicians and policy makers in the manuscript (Line 550-558).

13. Suggested Revision:

Conduct a thorough review for grammar and clarity, ensuring consistency in writing style and tone throughout the manuscript.

14.

Thank you, we have taken a careful look at the manuscript for grammar, clarity and consistency.

Additional Editor’s comments

1. I question the use of Google Scholar as a literature source. As you may know, most entries on the Google Scholar database are unmoderated and may not be legitimate research articles. It is also difficult to develop systematic search strategies for Google Scholar compared to dedicated academic database providers, such as OVID or EBSCOhost.

Thank you, we appreciate the difficulty in searching for studies through Google Scholar. However, a significant number of studies from Low and Middle-Income Countries (LMICs) are published in journals that are not indexed in the standard and well-moderated databases such as PubMed, EMBASE, LILACS and CINAHL, just to mention a few. These studies can be found in Google Scholar. Also, our review team has a lot of experience with searching Google Scholar. From our experience with systematic reviews from resource limited settings, a lot of non-indexed studies are usually missed when Google Scholar is excluded. Also, whilst it is true that many studies available in Google Scholar may not have undergone rigorous peer review or originate from credible journals and that they may not be legitimate research articles, unbiased systematic reviews should in principle include all (completed) studies but not only moderated studies. The aspect of the systematic reviews process that assesses quality in the included studies uses the same validated tools that ensure that all the included studies, regardless of whether they are published or not, are subjected to the same (unbiased) rigorous evaluation process. It should be noted that journals vary in robustness of the peer review process and that the fact that a paper has been peer-reviewed and published means it is of good quality. The Cochrane guidelines encourage the inclusion of unpublished studies in systematic reviews (to minimize publication bias). By incorporating studies identified through Google Scholar (not indexed in traditional databases), our approach is consistent with these guidelines and ensures a more comprehensive, unbiased and reliable review in line with a key tenet of systematic reviews, to attempt to locate and include all studies meeting pre-specified eligibility criteria. In view of this, we would like to maintain Google Scholar as a key database for this review.

2. Please concretely define whether you will compute OR or RR as the effect measure for dichotomous outcomes.

Thank you, we will use risk ratio (RR) as the effect measure for dichotomous outcomes and clarified this by removing odds ratio in the abstract (Lines 64) and main document (503-504).

3. Line 418, please remove capitalization on "Forest plots".

Capitalization on “Forest plots” removed.

4. Please do not use vague language such as "The synthesis may be stratified based on patient populations, components of interventions and study designs." Given that this is a prospective protocol, you should clearly define any subgroup or sensitivity analysis that you are going to conduct.

Thank you, we have deleted the sentence "The synthesis may be stratified based on patient populations, components of interventions and study designs" as the exact sources of heterogeneity and the anticipated sub-group analysis have been specified under the section “Heterogeneity and subgroup analysis” (Lines 511-512).

5. Note that you should not be using statistical measures of heterogeneity as the deciding factor for whether a random- or fixed-effects approach is used. This decision should be purely made based on characteristics of the included patients and studies -- in most cases, this will be a random-effects model unless you have a clear justification for using fixed-effects (e.g., the patient population are very similar, such as pooling studies from the same institution). Please concretely define your approach as a random-effects approach.

The sentence has been revised to indicate that a random-effects model will be used to pool studies (Line 67-68).

6. What are "appropriate imputation methods"? Clearly define them and any sensitivity analyses that will be used to assess the impact of imputation, and cite any methodological paper, if needed.

Thank you, missing data will not be imputed. Data will be analyzed using intention-to-treat (ITT) analysis. Where ITT analysis is not suitable, complete case analysis based on the number of patients for whom an outcome was recorded will be conducted. We have made the necessary revision (Line 504-506).

---

## [Editor Report · Decision Letter 1]

21 Apr 2025

Effectiveness of Care Coordination Interventions Delivered to Stroke Survivors in Low and Middle-Income Countries: Systematic Review and Meta-Analysis Protocol

PONE-D-24-49441R1

Dear Dr. Danso-Appiah,

We’re pleased to inform you that your manuscript has been judged scientifically suitable for publication and will be formally accepted for publication once it meets all outstanding technical requirements.

Kind regards,

Jiawen Deng

Academic Editor

PLOS ONE
---

## [Editor Report · Acceptance letter]

PONE-D-24-49441R1

PLOS ONE

Dear Dr. Danso-Appiah,

I'm pleased to inform you that your manuscript has been deemed suitable for publication in PLOS ONE. Congratulations! Your manuscript is now being handed over to our production team.

Kind regards,

on behalf of

Dr. Jiawen Deng

Academic Editor

PLOS ONE